# Positioning Performance of a Sub-Arc-Second Micro-Drive Rotary System

**DOI:** 10.3390/mi12091063

**Published:** 2021-08-31

**Authors:** Manzhi Yang, Zhenyang Lv, Chuanwei Zhang, Yizhi Yang, Gang Jing, Wei Guo, Zhengxiong Lu, Yumei Huang, Kaiyang Wei, Linyue Li, Bin Feng, Hongyu Ge

**Affiliations:** 1College of Mechanical Engineering, Xi’an University of Science and Technology, Xi’an 710054, China; Lp68813@sina.com (Z.L.); zhangcw@xust.edu.cn (C.Z.); jingang163@163.com (G.J.); Guow@xust.edu.cn (W.G.); 13296716754@163.com (Z.L.); ky15293595762@163.com (K.W.); li1424020163@126.com (L.L.); fengbinxust@xust.edu.cn (B.F.); ghy_xkd@sohu.com (H.G.); 2College of Humanities and Foreign Languages, Xi’an University of Science and Technology, Xi’an 710054, China; yyz195020246@sina.com; 3School of Mechanical and Precision Instrumental Engineering, Xi’an University of Technology, Xi’an 710048, China; hymxaut@163.com

**Keywords:** micro-drive rotary system, PZT (piezoelectric actuator), micro rotary mechanism, sub-arc-second, positioning performance, drive performance, transformation performance

## Abstract

In the macro/micro dual-drive rotary system, the micro-drive system compensates for the position error of the macro-drive system. To realize the sub-arc-second (i.e., level of 1″–0.1″) positioning of the macro/micro dual-drive rotary system, it is necessary to study the positioning performance of the sub-arc-second micro-drive rotary system. In this paper, we designed a sub-arc-second micro-drive rotary system consisting of a PZT (piezoelectric actuator) and a micro rotary mechanism, and used simulation and experimental methods to study the positioning performance of the system. First, the micro-drive rotary system was developed to provide ultra-precise rotary motion. In this system, the PZT has ultrahigh resolution at a level of 0.1 nanometers in linear motion; a micro rotating mechanism was designed according to the composite motion principle of the flexible hinge, which could transform the linear motion of piezoelectric ceramics into rotating motion accurately. Second, the drive performance was analyzed based on the drive performance experiment. Third, kinematics, simulation, and experiments were carried out to analyze the transformation performance of the system. Finally, the positioning performance equation of the system was established based on the two performance equations, and the maximum rotary displacements and positioning error of the system were calculated. The study results showed that the system can provide precision motion at the sub-arc-second and good linearity of motion. This study has a certain reference value in ultra-precision positioning and micromachining for research on rotary motion systems at the sub-arc-second level.

## 1. Introduction

Mechanical equipment manufacturing is critical for the manufacturing industry. As the precision requirements of mechanical systems in modern industry improve continually, the machining precision and motion precision requirements of equipment manufacturing of mechanical systems increase [1,2]. At present, precision and ultra-precision processing technologies are important methods for the realization of mechanical equipment precision manufacturing [3,4,5,6,7]. Precision feed technology is a vital method for the realization of ultra-precision processing technology [8,9,10]. Thus far, the macro/micro dual-drive system has resolved the contradiction between large motion travel and high motion precision, being able to provide motion with large travel and high precision, and the macro/micro dual-drive system has been used widely in high-technology fields such as the advanced war industry, biological medicine, and precision electronics [11,12,13,14,15,16]. In the macro/micro dual-drive system, the micro-drive system can compensate for the motion error of the macro-drive system and provide high-precision motion for the macro/micro dual-drive system [17,18,19,20].

In recent years, many scholars have mainly focused on mechanical structure design, structure optimization, dynamic analysis, and practical application of the micro-drive system. A precision MEMS (micro-electromechanical systems) mechanism with six degrees of freedom has been designed, and the mechanism can be applied to electron transport microscopes; in addition, the mechanism has realized precision rotary motion in three-dimensional space [21]. A micro bridge mechanism with three degrees of freedom has been developed, and the Z-direction of the mechanism has high stiffness and stability, meaning that it can resist a sizeable external thrust, and the structure optimization of the mechanism is completed by the sequential quadratic rule; the experimental results show that the optimization method is very effective [22]. A flexible micro mechanism with two-direction rotary motion driven by a giant magnetostrictive actuator has been proposed. The dynamic model of the mechanical parameters is given based on dynamic model establishment [23]. Xue et al. designed a new chip-level comb-drive magnified micro-drive platform with 3-DOF (degrees of freedom), and the maximum displacements of the platform in three directions were 25.2 μm × 20.4 μm × 58.5 μm [24]. Liu et al. presented a new compliant mechanism constituted by flexure hinges with two straight-circular and multi-notched components; the structural optimization of the mechanism was completed using a topological method, and the mechanisms’ performance was analyzed by finite element methods such as stiffness, rotary precision, and horizontal stress, with the results showing excellent mechanism performance [25].

At present, micro-drive systems research is concentrated mainly on linear motion, and a large amount of research has been used in production [26,27]. However, compared with linear systems, the research on micro-drive rotary systems is scarce and dated [28]. In addition, the positional accuracy of linear motion can reach the nanometer level [29,30], but the rotary motion is only at the arc-second (″) level [31,32,33,34].

The positional accuracy being restricted to the sub-arc-second level means that the maximum position error of the rotary motion system is limited to the scope of the 0.1 arc-second level. The rotary motion at the sub-arc-second level has essential applications in micromachining fields such as ultra-precision machining, chip processing, and ultra-precision medical instruments. The micro-drive rotary system provides a way to realize sub-arc-second positioning; therefore, a micro-drive rotary system with sub-arc-second positional accuracy has important value in the above micromachining fields. Thus, it is important to study the positioning performance of a sub-arc-second micro-drive rotary system, and this study can promote the development of ultra-precision micromachining technology.

As the positional accuracy of rotary motion is lower than linear motion and can only reach the level of arc-second, this paper presents research on the positioning performance of a sub-arc-second micro-drive rotary system, and this study can promote the development of micro-machining technology with ultra-precision (sub-arc-second) requirements. In this paper, a sub-arc-second micro-drive rotary system was developed to accurately transform linear motion with ultrahigh precision into rotary motion. The positioning performance of the system was analyzed based on the research on drive performance and transformation performance. The simulation and experimental results indicate that its motion accuracy can reach the sub-arc-second level, and the positioning performance was good.

The rest of this paper is organized as follows. Section 2 introduces the micro-drive rotary system. The drive performance analysis of the system by experiments is described in Section 3. In Section 4, three methods are used to analyze the performance of the system. In Section 5, the results of the positioning performance analysis are presented. The conclusions are presented in Section 6.

## 2. Micro-Drive Rotary System

At present, the positional accuracy of linear motion can reach the nanometer level, and the positional accuracy of a few micro-actuators can even reach the sub-nanometer level. However, the positional accuracy of rotary motion is only at the arc-second (″) level. In this paper, a micro-drive rotary system was designed to provide rotary motion at the sub-arc-second level. In this system, a new micro rotary mechanism can transform the linear motion into rotary motion with high precision, and the linear motion is driven by PZT with resolution at the sub-nanometer level.

The micro-drive rotary system is developed as shown in Figure 1: the fixed part and the gasket are designed to connect the two ends of the PZT to the micro rotary mechanism.

The motion of the micro-drive rotary system consists of the drive motion and the transformation motion. The drive motion of the micro rotary mechanism has a linear motion displacement ∆v, which is driven by the PZT, with driven voltage U. The transformation motion of the system has a rotary motion displacement ∆θ, which is transformed by the micro rotary mechanism with the linear motion displacement ∆v. Therefore, to analyze the positioning performance (i.e., the functional relationship between U and ∆θ) of the micro-drive rotary system, drive performance (i.e., the functional relationship between U and ∆v) and transformation performance (i.e., the functional relationship between ∆v and ∆θ) analyses are necessary.

### 2.1. Micro-Actuators

The two types of standard micro-actuators are PZT (piezoelectric ceramic actuator) and GMA (giant magnetostrictive actuator), and the motion precision of the PZT is higher than that of the GMA. PZT is a type of micro driver that functions according to the inverse piezoelectric effect of piezoelectric materials. PZTs can precisely drive the micro mechanism in the sub-10 micrometer range. Due to its advantages such as high precision, rapid response, small size, and large output force, PZTs have been used widely in the advanced war industry, biological medicine, precision electronics, intelligent robotics, and precision machinery. According to the parameter demands of this system such as the size of the driver, driving force, and driving displacement, the high-precision PZT of the P-235.1S model, produced by the PI Company, was selected. The PZT is shown in Figure 2. The main parameters of the PZT are shown in Table 1. Closed-loop travel was 0–15 μm, and the closed-loop resolution was 0.3 nm.

### 2.2. Micro Rotary Mechanism

When flexible material is stressed, the flexure hinge will produce minor deformation, which can realize the transmission motion and guiding motion according to the compound motion principle of the flexure hinge. In this paper, a micro rotary mechanism was designed based on the compound motion principle of the flexure hinge. The working principle of the micro rotary mechanism is shown in Figure 3. The micro rotary mechanism includes four parts: parts a, b, c, and d.

The structure and working principle of the micro rotary mechanism are shown in Figure 3. There are 26 flexure hinge elements in the x–y plane, with symmetrical distribution about center point 0, and the 26 flexure hinge elements are of the same size and are numbered, as shown in Figure 3. As the mechanism has a symmetrical structure when in motion, it is feasible only to study the parts in the top half of the mechanism (i.e., the above -x-axle parts of the mechanism) when analyzing its working condition. Due to the symmetrical structure, a parts contain a_1_ and a_2_, b parts contain b_1_ and b_2_, d parts contain d_1_ and d_2_, and part c is made up of 26 flexure hinges (1–26). The pivots of flexure hinges 2, 6, 8, 11, and 13 are connected with parts, and the a_1_ and a_2_ parts are fixed with 3 M4 socket head cap screws (the six screws are simplified for simple construction). The pivots of flexure hinges 3 and 5 are connected with d_1_ parts, and the d_1_ and d_2_ parts are connected with the rotary workbench with 3 M4 socket head cap screws (the six screws are simplified for simple construction). The pivots of flexure hinges 1, 4, 7, 9, 10, and 12 are connected with b_1_ parts. The connecting rod 123 (the connecting rod consists of points 1, 2, and 3, and the other connecting rods are defined in the same manner) completes the transformation of linear motion into rotary motion. The flexure hinges 6, 7, 8, 9, 10, 11, 12, and 13 complete the guidance for the mechanism, and the mechanism will have no non-motion direction (the *y*-axis is the motion direction) displacement. The flexure hinges 4 and 5 are used to balance the x-direction force and moment because the flexure hinge 1 moves around the pivot of flexure hinge 2 while the mechanism is working.

The mechanism is processed by wire-electrode cutting with spring steel 60Si2Mn material, which is shown in Figure 3b.

### 2.3. The Positioning Motion of the System

The positioning motion of the micro-drive rotary system consists of the drive motion and the transformation motion. Therefore, in order to realize high-precision rotary motion, it is essential to study the process of the two motions.

#### 2.3.1. The Drive Motion Process

The drive motion occurs when the system obtains a linear motion displacement Δv, which is driven by the PZT with driven voltage U (i.e., the drive process from U to Δv).

Piezoelectric ceramics are composed of large amounts of small crystal particles. The atoms of each small crystal particle are arrayed with a certain regularity, but the crystal lattices of each small crystal particle are arrayed in random order. As a result, the crystal particles of piezoelectric ceramics are arrayed in chaotic and random order, and the crystals with the above structure are known as polycrystalline structures. The distance between positive and negative charges inside piezoelectric ceramics will increase when increasing electric field intensity is applied to them. The electric field is in the same direction as the crystals’ polarization. Additionally, the length of the piezoelectric ceramics will enlarge along the direction of the crystals’ polarization. The process of transforming electrical energy into mechanical energy is known as the inverse piezoelectric effect.

When the electric field intensity of the inside of the PZT increases, the number of electric domains with the same direction as the electric field will decrease, and the distance among charges of piezoelectric ceramics will increase; then, the length of the piezoelectric ceramics will increase, and an elongation drive displacement of PZT is exported. In contrast, when the electric field intensity of the inside of the PZT decreases, then the length of the piezoelectric ceramics will shrink; therefore, a shortened drive displacement of PZT is exported.

Due to the material attributes of piezoelectric ceramics, PZTs have several disadvantages such as creeping, hysteresis, and nonlinear characteristics. These disadvantages are linked to the properties of piezoelectric materials such as the electrostriction, piezoelectric, inverse piezoelectric, and ferroelectric effects. In the micro-drive rotary system, the high-precision PZT of the P-235.1S model with a closed-loop control system was used. Thus, the PZT can avoid the above disadvantages, and as a result, the PZT can realize ultrahigh-precision positioning with the highest resolution of 0.3 nm.

#### 2.3.2. The Transformation Motion Process

In the system, the transformation motion is when the system transforms the linear motion displacement Δv into the rotary motion displacement Δθ according to a certain relationship based on the compound motion principle of the flexure hinge (i.e., the transformation process from U to Δv).

The micro rotary mechanism is a symmetrical structure with its center at point 0, as shown in Figure 3a; thus, it is feasible only to study the parts in the top half of the mechanism (i.e., the parts above the *x*-axis of the mechanism) when analyzing its working conditions. If the driven displacement of the system is Δv, the driven displacement of the top half of the mechanism is 0.5 Δv.

The transformation motion mainly relies on the flexure hinges 1, 2, 3, 4, and 5. In the motion of the system, the a, b, and d parts are considered a rigid body, the connecting rods 123 and 45 are also considered rigid bodies, and the flexure hinges 1, 2, 3, 4, and 5 are considered hinges.

## 3. The Drive Performance Analysis of the System

To study the drive performance of the system, a drive performance experiment was performed, as shown in Figure 4. The experimental apparatus consisted of the experiment box, the PZT, the micro rotary mechanism, the experiment base, two-sided displacement sensors (ZHONGYUAN MEASURING DGC-6PG/A measuring range: ±0.3 mm, total travel: 1~1.5 mm, linear error: ±0.5%, repeatability error: 0.05 μm), and two magnetic table mountains. To avoid external disturbance, the experiment was performed on a vibration isolation platform. In the experimental apparatus, the experiment base was fixed onto the experiment box, and the micro-drive system was fixed onto the experiment base. The driven voltage U was controlled by the control system of the PZT, and sensors 1 and 2 detected the displacement of the system in the direction of the *y*-axis, with values of δy_1_ and δy_2_. The linear motion displacement Δv is:(1)$$\Delta v=\left|\delta y_{1}\right|+\left|\delta y_{2}\right|$$

The drive performance of the system in the ascent stage (the stage with the driven voltage U increasing) and the descent stage (the stage with the driven voltage U decreasing) was tested, and the results of the experiment are shown in Figure 5a. In addition, the drive performance linear fit of the drive performance is shown in Figure 5b.

The linear equation fit between the driven voltage U and the linear motion displacement Δv of the system in the ascent stage is:(2)Δv=1.4438U−0.0467

The linearity of the linear equation is 0.9996.

The linear equation fit between the driven voltage U and the linear motion Δv of the system in the descent stage is:(3)Δv=1.4654U−0.2451

The linearity of the linear equation is 0.9977.

The drive performance equation can be analyzed from Equations (2) and (3) as:(4)$$\Delta v=\left\{\begin{array}{c} 1.4438U-0.0467\left(\mathrm{Ascent}\;\mathrm{stage}\right)\\ 1.4654U-0.2451\left(\mathrm{Descent}\;\mathrm{stage}\right) \end{array}\right.$$

The minimum linearity of the linear equation is 0.9977.

The drive error of the system δ_1_ can be calculated by the slopes of an ascent stage and descent stage through Equation (4), in other words,
(5)$$\delta_{1}=\frac{A_{2}-A_{1}}{A_{2}}\times 100\%=1.47\%$$
where *A*_1_ and *A*_2_ are the slopes of the ascent stage and descent stage of Equation (4).

Therefore, the drive performance Equation (4) was analyzed. The minimum linearity of the linear of the equation was 0.9967, and the drive error of the system (δ_1_) was 1.47%.

## 4. The Transformation Performance Analysis of the System

To study the drive performance of the system, three analysis methods were carried out: kinematics analysis, simulation analysis, and experimental analysis.

### 4.1. Kinematics Analysis

The transformation motion was analyzed by kinematics analysis, as shown below.

If the connecting rod 13 is considered a rigid body, then:(6)$$l_{13}=l_{1\prime 3\prime }$$
(7)$$l_{13}=\sqrt{{\left(R_{3}\cos\theta -x_{1}\right)}^{2}+{\left(R_{3}\sin\theta -y_{1}\right)}^{2}}$$
(8)$$l_{1\prime 3\prime }=\sqrt{{\left[R_{3}\cos\left(\theta +\Delta \theta \right){-x}_{1}\right]}^{2}+{\left[R_{3}\sin\left(\theta +\Delta \theta \right){-y}_{1}-\Delta v\right]}^{2}}$$

In Equations (7) and (8), (x_1_,y_1_) is the initial coordinate of point 1, θ is the included angle between L_03_ (point 2 is on the line), and R_3_ is the distance from point 3 to point 0 (R_3‘_ is also equal to the distance of point 3′ to point 0).

From Equations (6)–(8), then:(9)$$\left({2y}_{1}R_{3}+2\Delta {\mathrm{vR}}_{3}\right)\sin\left(\theta +\Delta \theta \right){+2x}_{1}R_{3}\cos\left(\theta +\Delta \theta \right)=\Delta v^{2}+2\Delta {\mathrm{vy}}_{1}{+2x}_{1}R_{3}{\cos\theta +2y}_{1}R_{3}\sin\theta$$

If $k_{1}{=2y}_{1}R_{3}+2\Delta {\mathrm{vR}}_{3}$, $k_{2}{=2x}_{1}R_{3}$, $k_{3}=\Delta v^{2}+2\Delta {\mathrm{vy}}_{1}{+2x}_{1}R_{3}{\cos\theta +2y}_{1}R_{3}\sin\theta$, and Equation (9) can be changed to:(10)$$k_{1}\sin\left(\theta +\Delta \theta \right){+k}_{2}\cos\left(\theta +\Delta \theta \right){=k}_{3}$$

Then:(11)$$\theta +\Delta \theta =\mathrm{Atan}2\left(k_{2}{,-k}_{1}\right)-\mathrm{Atan}2\left(k_{3},\pm \sqrt{k_{1}{}^{2}{+k}_{2}{}^{2}{-k}_{3}{}^{2}}\right)$$

Therefore:(12)$$\Delta \theta =\mathrm{Atan}2\left(k_{2}{,-k}_{1}\right)-\mathrm{Atan}2\left(k_{3},\pm \sqrt{k_{1}{}^{2}{+k}_{2}{}^{2}{-k}_{3}{}^{2}}\right)-\theta$$

When the system has different input linear displacements Δv driven by the PZT, the corresponding output angular displacements Δθ can be calculated using Equation (12). After placing the initial conditional parameters of the mechanism into Equation (12), the initial conditional parameters are as shown in Table 2.

Here, (x_1_, y_1_) is the coordinate value of point 1. (x_4_, y_4_) is the coordinate value of point 4. R2, R3, and R5 are the distance from point 2, point 3, and point 5 to the origin O, respectively. θ_2_, θ_3_, and θ_5_ are the angles of the lines from point 2, point 3, and point 5 to the origin O and the -x-axle, respectively.

Equation (12) is the theory model equation of transformation performance.

### 4.2. Simulation Analysis

When the transformation performance of the system was analyzed using simulation, the transformation relationship between Δv and Δθ was calculated using the Statics module of ANSYS Workbench.

The 3D model of the micro rotary mechanism was imported into the finite element software. The radius and minimum thickness values of the flexure hinges were 3 mm and 1 mm, and the dimensions of the mechanism were 160 mm × 150 mm × 50 mm (length × width × height). The parameters of 60Si2Mn were chosen as the material properties of the mechanism. To conveniently load the mechanism, the imprint face in the mechanism was constructed, and the structure of the imprint face was fitted to the inner surface between the mechanism and the PZT.

When meshing, the meshing of the entire mechanism was completed using free meshing with 1 mm. Then, refinement grid cells were used for 52 cylinders of flexure hinges with 0.5 mm mesh. After meshing, the number of nodes in the model was 1,590,075, and the number of meshes was 938,699. The meshed model is shown in Figure 6: the mesh of the arc portion of the flexure hinge was finely divided, and there was no breakage, indicating that the meshing quality was good.

When loading, fixed constraints were placed on the six threaded hole cylinders of M4 in the parts. The loading conditions were the y positive direction and y negative direction displacements with 0.5Δv, which were loaded at the position of the imprint face. The preparation for the ANSYS Workbench analysis is illustrated in Figure 6.

In the mechanism, the 3D coordinates of the lower surface center point and upper surface center point were defined as (0, 0, 0) and (0, 0, 50). The x-direction displacements of the points were calculated with coordinates (0, 63.5, 50) and (0, −63.5, 50) by the probe method, after which the Δθ of the system could be analyzed. For example, when Δv = 14.96 μm was loaded, the x-direction displacements of the points with coordinates (0, 63.5, 50) and (0, −63.5, 50) were 3.23 μm and −3.30 μm, and Δθ = arctan(0.00653/127) = 10.60″. As a result, when the system input linear displacement was 14.96 μm, the output rotary displacement was 10.60″. In the same way, the output rotary displacements Δθ corresponding to different input linear displacements Δv of the system were calculated. The calculation process is shown in Figure 7.

The simulation results show that the system can transform linear motion (Δv) into rotary motion (Δθ), and they verify the validity of the theory model Equation (12) of transformation performance.

### 4.3. Experimental Analysis

To analyze the transformation performance of the system, the transformation performance experiment was performed. The transformation performance experiment is shown in Figure 8. The experimental apparatus consisted of the experiment box, the PZT, the micro rotary mechanism, the experiment base, two-sided displacement sensors (ZHONGYUAN MEASURING DGC-6PG/A measuring range: ±0.3 mm, total travel: 1~1.5 mm, linear error: ±0.5%, repeatability error: 0.05 μm), two straight displacement sensors (ZHONGYUAN MEASURING DGC-8ZG/C measuring range: ±0.6 mm, total travel: 3 mm, linear error: ±0.5%, repeatability error: 0.03 μm), and four magnetic table mountains. To avoid external disturbance, the experiment was performed on a vibration isolation platform. In the experimental apparatus, the experiment base was fixed onto the experiment box, and the micro-drive system was fixed onto the experiment base.

The linear input displacement △v of the system was determined by calculating the y-direction displacement of the b parts with side sensor 1 (with a value of δy_1_) and side sensor 2 (with a value of δy_2_). The output angular displacement Δθ of the system was indirectly determined by calculating the x-direction displacement of the d parts with straight sensor 3 (with a value of δx_1_) and straight sensor 4 (with a value of δx_2_). The experimental results of △v and Δθ were calculated as Equations (13) and (14).
(13)$$\Delta v=\;\left|{\delta y}_{1}\right|+\left|{\delta y}_{2}\right|$$
(14)$${\Delta \theta =\tan}^{-1}\left(\frac{\left|{\delta x}_{1}\right|\left|{\delta x}_{2}\right|}{l}\right){=\tan}^{-1}\left(\frac{\left|{\delta x}_{1}\right|\left|{\delta x}_{2}\right|}{12,700}\right)$$

In the experiment, to analyze the transformation performance accurately, 10 groups with Δv and Δθ were detected by different driven voltages of the PZT.

### 4.4. Comparison and Discussion

To analyze the transformation performance accurately, three methods were used in this paper: the theoretical analysis of theory model Equation (12) of transformation performance in Section 4.1, the simulation analysis in Section 4.2, and the experimental analysis. The results of the above three methods in analyzing the relationships between Δv and Δθ were contradictory, as shown in Figure 9.

The transformation performance linear fit of the drive performance is shown in Figure 10.

The linear equation fit of the theory analysis is:Δθ = 0.6157Δv + 0.0003(15)

The linearity of the linear equation is 1.

The linear equation fit of the simulation analysis is:Δθ = 0.7083Δv + 0.0005(16)

The linearity of the linear equation is 1.

The linear equation fit of the experimental analysis is:Δθ = 0.6656Δv − 0.5886(17)

The linearity of the linear equation is 0.9997.

Due to model simplification error and mechanical motion error, the analytical results of FEM are different from those of the theory analysis and experimental analysis.

For engineering application convenience (for example, the programming in CNC), the coefficient of Δv in Equations (15)–(17) was averaged because of its good linearity, and the transformation motion of the system can be described as in Equation (18).
Δθ = 0.6632Δv(18)

The resolution of the angular motion of the system can be calculated from the resolution of the piezoelectric ceramics in Table 1 (converted from nanometer to micron) combined with Equation (18): 0.0003×0.6332=0.00019″. Therefore, the resolution of the angular motion of the system is 0.00019″.

According to the results of the transformation performance analysis, the following conclusions can be made:

(1) The transformation motion can be described as in Equation (18), which can be conveniently applied in practical engineering problems.

(2) The transformation maximum error of the system (δ_2_) is 7.50%. From Equations (15)–(17), the error between the theoretical analysis and the experiment analysis is 7.5131% (calculated by the slopes of Equations (15) and (17)), and the error between the simulation analysis and experimental analysis is 6.42% (calculated by the slopes of Equations (16) and (17)); therefore, the maximum error (δ_2_) of the transformation motion is 7.50%.

(3) The minimum linearity of the transformation performance Equation (18) is 0.9997.

## 5. The Positioning Performance Analysis of the System

For the positioning motion of the micro rotary system, the linear displacement Δv is driven by the PZT with the driven voltage U based on the drive motion, and the rotary displacement Δθ is transformed by the micro rotary mechanism with the linear input displacement Δv based on the transformation motion. The positioning performance of the micro rotary system is focused on the relationship between the driven voltage U and the rotary displacement Δθ (i.e., the functional relationship between Δv and Δθ).

The drive motion equation of the system in Equation (4), the transformation motion equation of the system in Equation (18), and the positioning motion equation of the system can be analyzed as:(19)$$\Delta \theta =\left\{\begin{array}{c} 0.9575U-0.0310\left(\mathrm{Ascent}\;\mathrm{stage}\right)\\ 0.9719U-0.1626\left(\mathrm{Descent}\;\mathrm{stage}\right) \end{array}\right.$$

The minimum linearity of Equations (4) and (18) is 0.9977 and 0.9997. Therefore, the minimum linearity of Equation (19) is 0.9974.

The positioning motion is expressed as Equation (19).

If the positioning error of the system is δ, then:(20)$$\delta =1-\left({1-\delta }_{1}\right)\left({1-\delta }_{2}\right)$$

Placing δ_1_ = 1.47% and δ_2_ = 7.50% into Equation (20), the positioning error of the system is δ = 8.86%.

Inserting the maximum driven voltage U_max_ = 10 V and the positioning error δ = 8.86% into Equation (20), the maximum rotary motion displacement matrix A is:(21)$$A=\left[\begin{array}{cc} {\Delta \theta }_{1\max} & e_{1\max}\\ {\Delta \theta }_{2\max} & e_{2\max} \end{array}\right]\left[\begin{array}{cc} 9.54^{\prime \prime } & 0.82^{\prime \prime }\\ 9.56^{\prime \prime } & 0.85^{\prime \prime } \end{array}\right]$$

In Equation (21), Δθ_1max_ and Δθ_2max_ are the maximum output rotary displacements in the ascent and descent stages, and e_1max_ and e_2max_ are the maximum positioning error values in the ascent and descent stages.

The maximum rotary motion displacement of the system is shown in Figure 11.

According to the results of the positioning performance analysis, the following conclusions can be made.

(1) The positioning motion can be expressed as Equation (19), in which the minimum linearity is 0.9974.

(2) The system can provide precision motion at the sub-arc-second level as the maximum positioning error value of the system is 0.85″.

(3) The maximum output rotary displacement of the system is 9.56″ with a positioning error of 8.86%, so the positioning range of the system is approximately 0″–9.56″.

(4) In comparison with other studies, the positioning accuracy achieved in this study is higher and can reach the sub-arc-second level, as shown in Table 3.

## 6. Conclusions

This paper proposes a micro-drive rotary system that can transform linear motion into rotary motion with ultrahigh precision. Meanwhile, the positioning performance of the system was analyzed based on the performance analysis of drive motion and transformation motion. The simulation and experimental results show that the positioning accuracy can reach the sub-arc-second level and the positioning performance is excellent.

By transforming linear motion into rotary motion, the system can provide precision motion at the sub-arc-second level, with a maximum positioning error value of 0.85″. The positioning motion Equation (20) is given, the minimum linearity of which is 0.9974, and the maximum output rotary displacement of the system is 9.56″ with a positioning error of 8.86%. The results indicate that the position performance of the system is excellent.

The sub-arc-second micro-drive rotary system can independently provide ultrahigh rotary precision motion, and can also compensate for the motion error of the macro-drive rotary system to form a large-scale sub-arc-second macro/micro dual-drive rotary system. This research is beneficial to the study of rotary motion with sub-arc-second positioning precision, and can help to promote the development of macro/micro dual-drive technology. Future work will be carried out on the all-load-bearing macro/micro dual-drive rotary systems with sub-arc-second motion accuracy.

## Figures and Tables

**Figure 1 micromachines-12-01063-f001:**
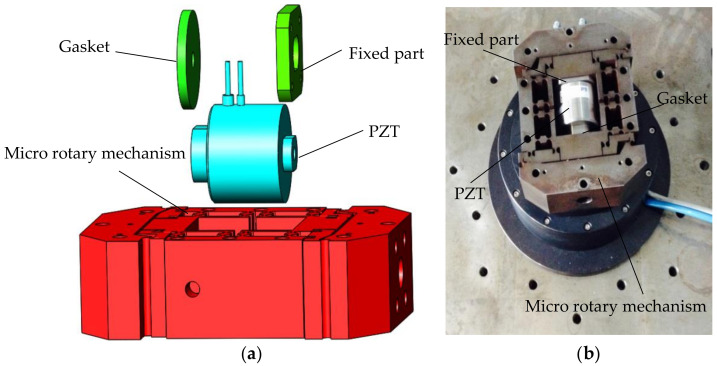
The micro-drive rotary system. (**a**) Explosion view of the micro-drive rotary system; (**b**) Image of the micro-drive rotary system.

**Figure 2 micromachines-12-01063-f002:**
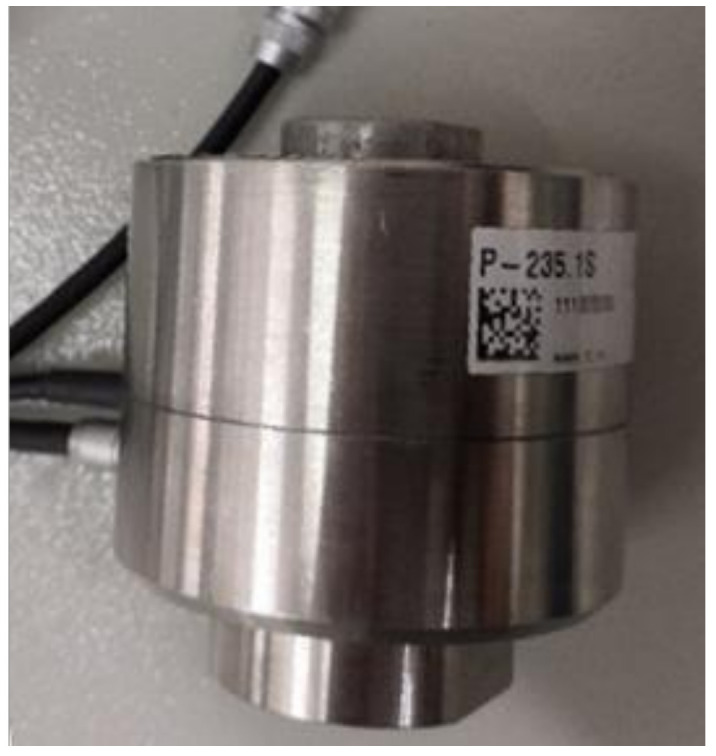
The picture of the P235.1S PZT.

**Figure 3 micromachines-12-01063-f003:**
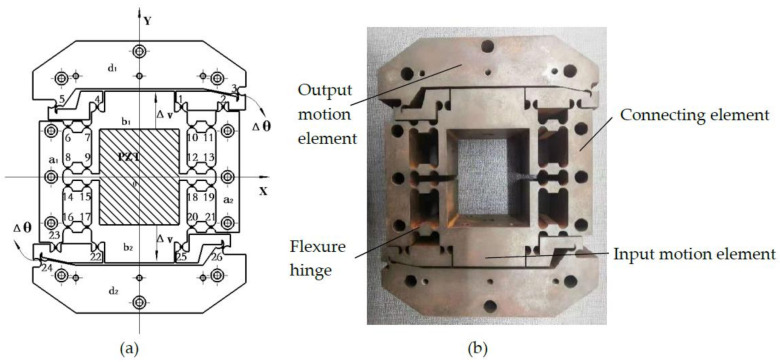
The working principle of the micro rotary mechanism. (**a**) Schematic diagram of micro rotary mechanism. (**b**) Image of the micro rotary mechanism.

**Figure 4 micromachines-12-01063-f004:**
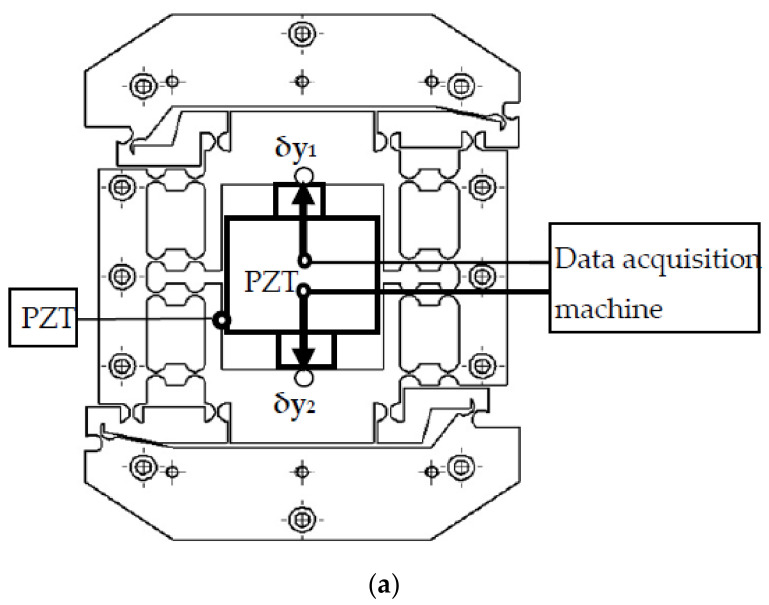
The drive performance experiment. (**a**) Schematic diagram of the drive performance experiment. (**b**) Image of the drive performance experiment.

**Figure 5 micromachines-12-01063-f005:**
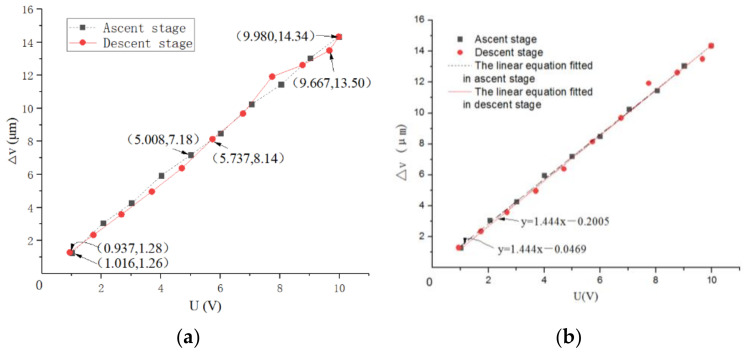
(**a**) The results of the experiment. (**b**) Linear fit of the drive performance.

**Figure 6 micromachines-12-01063-f006:**
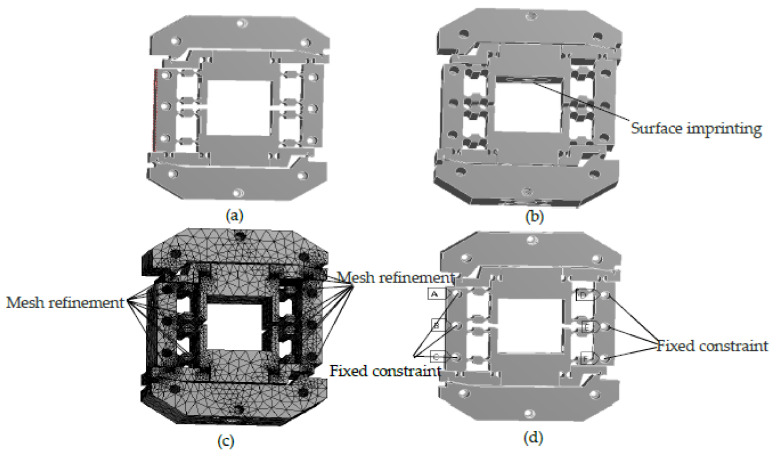
The preparation for ANSYS Workbench analysis. (**a**) Importing model; (**b**) Adding surface imprinting; (**c**) Meshing; (**d**) Constraint condition.

**Figure 7 micromachines-12-01063-f007:**
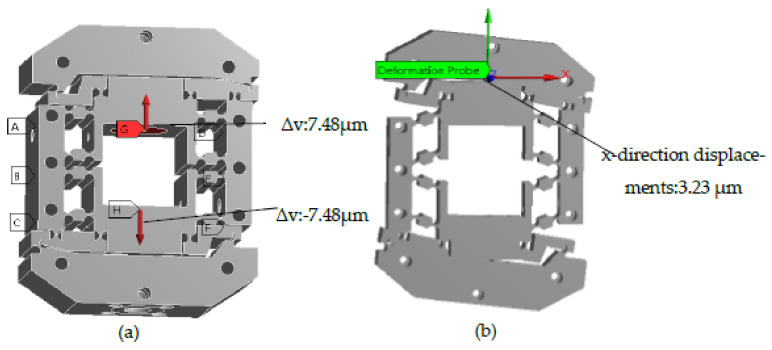
The maximum simulation displacements of the micro rotary mechanism. (**a**) Loading displacements. (**b**) Result of simulation.

**Figure 8 micromachines-12-01063-f008:**
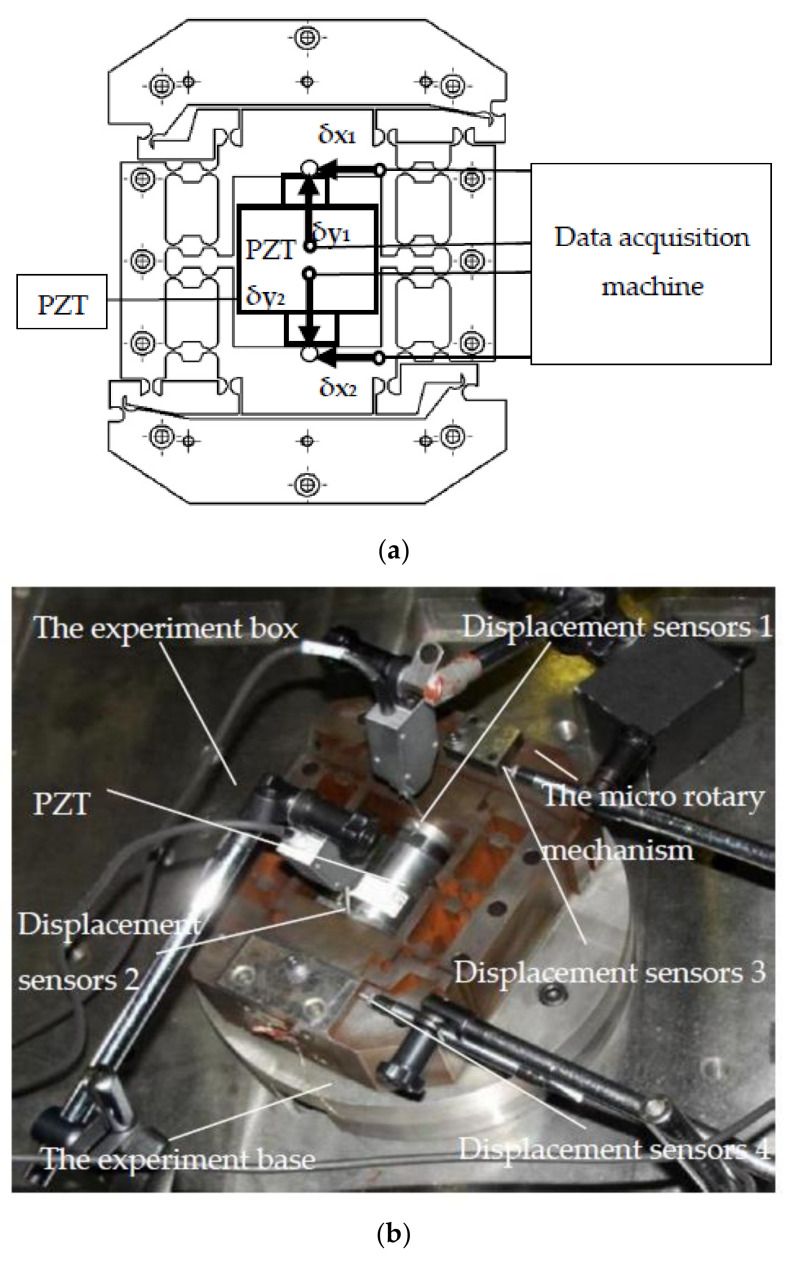
The transformation performance experiment. (**a**) Experimental schematic diagram of the drive performance experiment. (**b**) Image of the experiment.

**Figure 9 micromachines-12-01063-f009:**
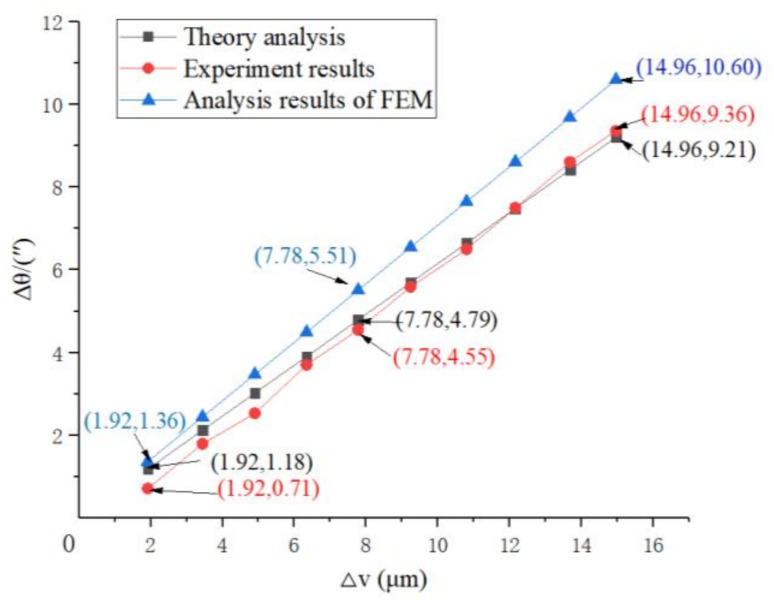
The results of the above three methods in analyzing the relationships between Δv and Δθ.

**Figure 10 micromachines-12-01063-f010:**
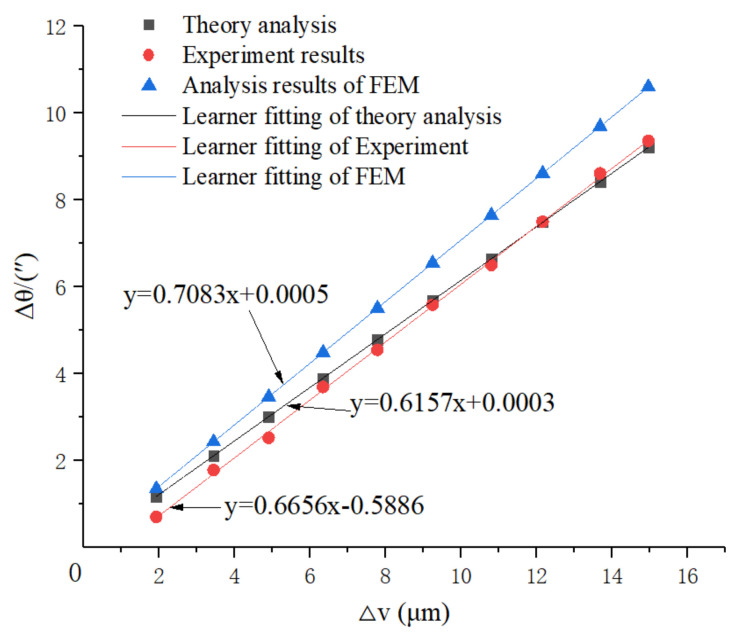
The transformation performance linear fit of the drive performance.

**Figure 11 micromachines-12-01063-f011:**
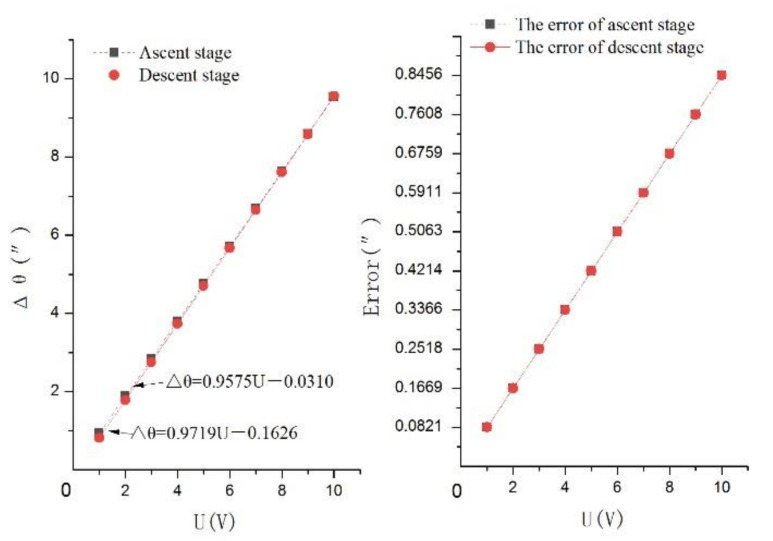
The maximum rotary motion displacement of the system.

**Table 1 micromachines-12-01063-t001:** Main parameters of the P235.1S PZT.

Index	Parameter
Length	55 mm
Cylinder diameter	49.8 mm
Closed-loop travel	0–15 μm
Resolution	0.3 nm
Linearity	0.2%
Static large-signal stiffness	860 N/m
Operating voltage	0–10 V
Maximum push	30,000 N
Maximum pull	3500 N
Shear force limit	707 N
Torque limit	2 Nm
Maximum motion frequency	300 HZ

**Table 2 micromachines-12-01063-t002:** The initial conditional parameters.

Parameter	x_1_	y_1_	x_4_	y_4_	R_2_	θ_2_	R_3_	θ_3_	R_5_	θ_5_
value	26.5	44.0	−26.5	44.0	68.8	40.0	81.0	40.0	68.8	140.0

**Table 3 micromachines-12-01063-t003:** This design compared to other precision rotary motion systems.

References	Author	Year	Positioning Error Value (″)
[9]	Yang, W.J. et al.	2021	2.98
[11]	Graser Philipp et al.	2021	1.3(72.5 μrad)
[32]	Liu, W.; Li, X. et al.	2018	7.2
[33]	Lorenzo, Iafolla. et al.	2021	10
[34]	Mou, J.P.; Su, J.T.; Huang, T.C.	2021	4
This paper	Yang M.Z.et al.	2021	0.85

## Data Availability

The data presented in this study are available on request from the corresponding author.

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
