# Peer review of "Positioning Performance of a Sub-Arc-Second Micro-Drive Rotary System"

_micromachines, 2021, doi:10.3390/mi12091063_

Round 1

Reviewer 1 Report

Please read the attached file. 

Author Response

Dear Reviewer,

Thank you for your comments on our manuscript titled “Positioning Performance of a Sub-Arc-Second Micro-Drive Rotary System”(micromachines-1289008). Those comments are all valuable and very helpful for revising and improving our paper, as well as the important guiding significance to our researches. We have studied comments carefully and have made correction which we hope meet with approval. Revised portion are marked in red in the paper. 

We appreciate for your warm work earnestly, and hope that the correction will meet with approval.

Once again, thank you very much for your comments and suggestions.

  Yours sincerely,

                                                                                                                       Manzhi Yang

                                                                         Corresponding author                                                                   E-mail: [email protected]

Reviewer 2 Report

The authors describe the analysis on the micro-drive rotary system which reaches reference positioning accuracy considering other rotary motion systems. The following remarks and questions should be addressed:

L167-172 – Change Figure 1 to Figure 3 in the text, reduce the repetitiveness of this paragraph

L168 – where are ‘c’ parts located? There is no description in Fig 3 as well as in the text.

L292 – k1 and k2 were already used as slope coefficients in Eq. 5

L300 – Table 2  - what is the meaning of X2, Y2, R1, R2 and θ1,2,3 concerning the equations before it?

L335 – Figure 7 doesn’t depict stress – change the caption.

L339 – Experimental

L341 – ‘The experimental is shown in Figure 8’ – the sentence lacks a noun.

L340-346 are repeated after L 243-250

Eq. 14 – How this corresponds with values given in L329? Please check the equation 14 and L329.

L425 – If the positioning error is up to 8.86% which translates to approx. 0.85’’, can we write the range of the system as exactly 0’’ – 9.56’’?

Also, the authors contradict themselves by stating: “The positional accuracy on the sub-arc-second level means the maximum position error of the rotary motion system has controlled by the scope of 0.1 arc-second level” in L76-77 – 0.85 > 0.1. Please comment and/or rephrase those sentences.

How does this design compare to other precision rotary motion systems? The authors could prepare a tabular comparison to conclude on the reference of their achievement.

Many language and style mistakes, as well as the general repetitiveness of some paragraphs, make this paper hard to follow. Proofreading by a native speaker is advised.

Author Response

(The authors gave the same response as above.)

Reviewer 3 Report

Positioning Performance of a Sub-arc-second Micro-drive Rotary System

Manzhi Yang1,Zhenyang Lv, Yizhi Yang, Gang Jing, Wei Guo, Yumei Huang, Kaiyang Wei, Linyue Li, Bin Feng1 and Hongyu Ge

The topics and scope of the articles comply with the requirements of the journal. The paper presents an interesting design of a mechanical system that converts linear motion into rotary motion. The design is intended for use with regard to a small movement performed by a piezoelectric stack - approx. 15 um converted into a precise rotational movement in the range of approx. 9 ", but with a very high accuracy. The resulting accuracy of the rotational movement is considered to be the main achievement. The system has been experimentally verified.

The text is well formatted, all the proper chapters (sections) are there. The theoretical section is well described. The analysis are properly explained. The error values are appropriately calculated. The simulations are properly carried out. All the important information (such as the limits of the systems are well described).

I highly value the substantive status of the presented work and recommend it for publication, but with the need to introduce minor corrections.

My comments are as follow.

Comment 1

Line 55 10 μm× 20 μm× 0.2 μm in size

Comment 2

Line 73 Please consider to reorganise

Comment 3

Line 141 Please explain the travel of 15um under 10 V suggested in  Tab.1.  

Comment 4

Line 231  Please explain why system is designed as symmetrical structure.   It may work as a half of it. It would “safe” a motion of PZT.   

Comment 5

Line 256 (a) Experimental … (b) …….  ???

Comment 6

Line 264 (b) The drive performance linear fit of the drive performance.

Comment 7

Line 372 Please explain the difference between the FEM results and the other two – Figure 9 and 10

Comment 8

Line 372 Please explain why the experimental characteristics were obtained only by increasing the argument - avoiding possible differences when decreasing the value of the argument. Argument u and U

Comment 9

Line 425  The designation of the operating voltage with the letter U is unfortunate when the linear movement is denoted by the letter u.

Comment 10

Line 430   …the results show that the linearity of the system motion is excellent. – not acceptable especially in the conclusion.

Comment 11

Line 437 Tab. 3. It is recommended to add the column with positioning range.

For the future work.

It would be interesting to find the maximum torque the system can handle in a rotating motion.

Author Response

Dear Reviewer,

Thank you for your comments on our manuscript entitled " Positioning Performance of a Sub-Arc-Second Micro-Drive Rotary System " (micromachines-1289008).These opinions have important guiding significance to the modification and perfection of the thesis.We have carefully studied the comments and have made corrections, which are in the attachment and hope to be approved.

We appreciate for your warm work earnestly, and hope that the correction will meet with approval.

Once again, thank you very much for your comments and suggestions.

  Yours sincerely,

                                                                                                                       Manzhi Yang
